# Nutritional Status and Feeding Difficulty of Older People Residing in Nursing Homes: A Cross-Sectional Observational Study

**DOI:** 10.3390/nu17162607

**Published:** 2025-08-11

**Authors:** Hansen (Cindy) Tang, Kazem Razaghi, Wenpeng You, Yu (Carrie) Cheng, Lei (Tina) Sun, Ivy Wong, Hui-Chen (Rita) Chang

**Affiliations:** 1School of Nursing and Midwifery, Parramatta South Campus, Western Sydney University, Parramatta, NSW 2151, Australia; 22154560@student.westernsydney.edu.au (H.T.); k.razaghi@westernsydney.edu.au (K.R.); w.you@westernsydney.edu.au (W.Y.); 2School of Nursing, University of Wollongong, Wollongong, NSW 2522, Australia; carriecheng632@gmail.com; 3Chinese Australian Services Society (CASS), Residential Aged Care Facility, Campsie, NSW 2194, Australia; tina_sun@cass.org.au (L.S.); ivy_wong@cass.org.au (I.W.)

**Keywords:** nutritional status, dementia, nursing home, cognitive impairments, feeding difficulties

## Abstract

**Aims:** To investigate the nutritional status and feeding behaviours of nursing home residents and the impact of cognitive impairments and feeding difficulties on nutritional health. **Design:** A cross-sectional observational design was employed. **Methods:** The study assessed 51 nursing home residents using the Mini Nutritional Assessment Short-Form (MNA-SF) for nutritional status, the Feeding Difficulty Index (FDI) for mealtime behaviours, and the MoCA (Montreal Cognitive Assessment or The MoCA Test) for cognitive function. **Results:** The average age of participants was 87.8 years. Nearly half (47.1%) were at high risk of malnutrition, and 13.7% were classified as malnourished. The average MoCA score was 14, indicating moderate cognitive impairment, which was inversely associated with nutritional status. Feeding difficulties were common, as follows: 74.5% of residents paused feeding for over one minute, and 62.8% were distracted during meals. A longer duration of nursing home residency was associated with poorer nutritional outcomes. Overall, 65% of residents required mealtime assistance, with higher FDI scores correlating with greater support needs. Significant positive correlations were found between cognitive function and nutritional status (r = 0.401, *p* = 0.037) and between food intake and nutritional status (r = 0.392, *p* = 0.004). In contrast, residency duration (r = −0.292, *p* = 0.037) and feeding difficulties (r = −0.630, *p* < 0.001) were negatively associated with MNA-SF scores. FDI scores were strongly associated with the level of assistance required during meals (r = 0.763, *p* < 0.001). This study highlights the critical need for targeted nutritional assessments and interventions in nursing homes, especially for residents with dementia facing cognitive impairments and feeding difficulties. Enhancing staff training on recognising and addressing eating challenges and risk factors is essential for improving nutritional well-being. **Conclusions:** The study highlighted the profound impact of cognitive impairments and feeding difficulties on the nutritional health of nursing home residents, indicating a high prevalence of malnutrition and a need for comprehensive mealtime assistance.

## 1. Introduction

The global ageing population presents significant health challenges, with nutritional health among older people in nursing home emerging as a critical concern. The World Health Organisation (2022) projects a doubling of the population aged 60 years and older by 2050, underscoring the urgency of addressing nutritional health to improve quality of life among older people [1]. Ageing is associated with physiological changes that affect nutritional status and dietary needs, increasing the risk of malnutrition and related health complications [2,3,4]. Given these implications, nutritional health within nursing homes emerges as a paramount concern for ensuring positive health outcomes among the older people. Brglez et al. (2022) emphasise the health concerns associated with malnutrition among older populations, advocating for a more nuanced understanding and approach to nutritional care in these settings [5]. The complexity of this issue is underscored by the findings of Bassola et al. (2020), who highlighted the importance of assessing the nutritional status of the older people and the need for enhanced nurse education on malnutrition in this demographic to improve care outcomes [6].

## 2. Background

Despite the critical role of nutrition in the health of older people, research specifically targeting meal relevance and nutritional assessments within nursing homes remains scarce. Malnutrition among older people in nursing homes is a well-documented issue, with studies indicating a high prevalence of malnutrition and its severe consequences on health and well-being [7,8,9,10,11,12]. The impact of malnutrition includes a decline in quality of life, increased disease incidence, and higher mortality rates, highlighting the need for comprehensive nutritional care and support [13]. This malnutrition crisis is further compounded by inadequate attention to the specific nutritional needs and feeding behaviours of this population, signifying a profound gap in the current understanding of their feeding behaviours, challenges, and preferences. Corcoran et al. (2019) and Lin et al. (2023) underscore the need for more focused research in this area, particularly highlighting the scant examination of feeding behaviours and the crucial role of nursing staff in mealtime facilitation [8,14].

In this study, the term feeding difficulty is used to refer to observable mealtime challenges, as assessed by the Feeding Difficulty Index (FDI). The FDI captures a range of behaviours that reflect functional impairments during feeding, including physical resistance to feeding, inappropriate verbal or physical responses, lack of attention or engagement, motor difficulties with food handling, and oral-motor issues such as dribbling or prolonged chewing [15,16,17]. While the broader term feeding behaviours can encompass food preferences, habits, and attitudes, this study specifically focuses on mealtime functional behaviours that interfere with effective food intake.

The significance of nutrition for older people cannot be overstated, yet research on the dietary relevance and nutritional assessments in nursing homes remains sparse, underscoring a notable research gap in addressing the nutritional well-being of residents. While existing studies predominantly focus on the nutritional status of the older people in hospital and acute care environments [14], where only 1.2% of older people were found to have a normal nutritional status while 72.1% were malnourished, nursing homes have received comparatively less attention. Given that nursing homes serve as the primary residence for many frail older adults, understanding nutrition in this setting is crucial for long-term care planning and interventions. Against this backdrop, the Australia new Aged Care Quality Standards (ACQSC, 2024) particularly Standard 6: Food and Nutrition, play a crucial role in setting benchmarks for nutritional care. This standard underscores the importance of providing nutrient-dense meals that are rich in high biological value proteins, healthy fats, and essential micronutrients such as vitamin D, calcium, and B vitamins. It also advocates for culturally appropriate food options tailored to residents’ diverse backgrounds, as well as the provision of texture-modified diets (e.g., pureed foods) for those experiencing dysphagia. In addition, the standard emphasises the psychosocial aspects of the dining environment, including mealtime assistance, an inviting and supportive atmosphere, and visually appealing meal presentation, all of which are associated with improved intake among older adults [18]. At the same time, Australian nutrition experts recommend a combination of oral nutritional supplements (ONS), micronutrient supplementation, and enteral nutrition to optimise nutritional outcomes and uphold person-centred care in this population [19]. These strategies advocate for a more nuanced understanding and approach to nutritional care in nursing home, aiming to ensure that the dietary needs of residents are met in a manner that promotes their health, dignity, and quality of life.

Older adults with cognitive impairments especially those living with dementia are at increased risks of malnutrition due to their age-related malnutrition syndrome multifactorial condition characterised by executive dysfunction, disorientation, memory loss leading to missed meals, swallowing difficulties, reduced appetite, and resistance during mealtimes [20]. These cognitive issues often manifest as feeding difficulties, making it harder for individuals to initiate or complete meals independently. Feeding difficulties, which include observable mealtime challenges like food refusal, distraction, and the need for assistance, have been shown to directly impact food intake and nutritional status [21]. The existing literature underscores that these factors often coexist and synergistically increase the risk of malnutrition. By specifically examining the interplay between cognitive decline, nutritional status, and feeding difficulties, this study aims to investigate the nutritional status and feeding behaviours of nursing home residents and the impact of cognitive impairments and feeding difficulties on nutritional health.

## 3. Methodology

### 3.1. Design

This study employed a cross-sectional observational design, adhering to the Strengthening the Reporting of Observational studies in Epidemiology (STROBE) guidelines [22] to ensure the comprehensive and systematic examination of nutritional status, feeding behaviours, and cognitive function among nursing home residents. This study followed the STROBE guidelines for reporting cross-sectional studies.

### 3.2. Study Setting and Sampling

This study was conducted in a purposively selected nursing home in New South Wales, Australia, comprising two care units that agreed to participate. These units were selected to ensure geographic diversity and demographic variability among residents, which contributed to the contextual relevance of the study. A total of 51 residents were included in the final sample.

A priori power analysis using G*Power version 3.0.10 indicated that 57 participants would be required to achieve 95% statistical power at a significance level (α) of 0.05, assuming a medium effect size (Cohen’s d = 0.40). Ultimately, 51 residents were recruited. The primary reasons for the sample size were practical constraints, including limited availability of eligible residents, the relatively small size of the participating units, and strict exclusion criteria. Although it did not reach the expected effect size, the study identified significant associations between cognitive impairment, feeding difficulties, and malnutrition risk, highlighting critical gaps in mealtime care for older resident with dementia. Notably, previous study investigating similar topics have employed comparable sample size and were also able to identify statistically significant findings [23].

Inclusion criteria were as follows: (1) residents aged 65 years or older; and (2) those able to participate in mealtime observations, with or without verbal communication. Exclusion criteria included the following: (1) residents with medical conditions that precluded oral intake (e.g., tube feeding); (2) those receiving end-of-life palliative care; and (3) those unable to consent due to severe cognitive impairment and those whose legal guardian declined participation.

### 3.3. Study Instrument

Data were collected using validated instruments known for their reliability and validity in assessing nutritional status and feeding behaviours in older populations. These included the Feeding Difficulty Index (FDI) to evaluate challenges encountered during mealtime, the MoCA (Montreal Cognitive Assessment or The MoCA Test) for cognitive function, and the Mini Nutritional Assessment Short-Form (MNA-SF) to determine the nutritional status of the participants.

The Feeding Difficulty Index (FDI), originally developed by Liu et al. (2015) [16], was designed to identify feeding challenges among older Chinese adults with dementia who often require assistance during meals. It consists of 19 items across the following four domains: distractibility or condition, difficulty in accessing food, food refusal, and motor difficulties. Each item is scored on a 3-point scale (0 = never, 1 = sometimes, 2 = often), yielding a total score ranging from 0 to 57, with higher scores indicating greater feeding difficulty [16].

This approach is supported by prior research demonstrating the FDI’s utility in diverse long-term care settings [24]. The FDI’s content validity was established with item-level Content Validity Index (CVI) values exceeding 0.80, reviewed by a panel of eight experts in neurology, psychiatry, gerontology, and rehabilitation. Inter-rater reliability and test–retest reliability were both reported as high in the original validation study [16].

The Montreal Cognitive Assessment (MoCA) is a brief screening tool for detecting mild cognitive impairment and early-stage dementia. It evaluates various cognitive domains, including attention and concentration, executive functions, memory, language, vasoconstriction skills, conceptual thinking, calculations, and orientation. The total MoCA score ranges from 0 to 30, with higher scores indicating better cognitive functioning. A score below 26 is generally considered indicative of cognitive impairment [25]. The MoCA has demonstrated high sensitivity and specificity in older populations and has been validated in several languages and cultural settings.

The Mini Nutritional Assessment Short-Form (MNA-SF) streamlines the original Mini Nutritional Assessment for swift and user-friendly evaluation of nutritional status in older people. This abbreviated form, highly regarded for its simplicity and effectiveness, encompasses the following six key areas: dietary intake, recent weight loss, mobility, psychological stress or acute disease, neuropsychological issues, and either Body Mass Index (BMI) or calf circumference [26]. Scoring for each of these items ranges from 0 to 3. Upon summing these scores, individuals are categorised into one of the following three nutritional states: normal (12–14 points), at risk of malnutrition (8–11 points), or malnourished (0–7 points), as detailed by Kaiser et al. (2009) [26]. The MNA-SF demonstrates content validity through its comprehensive coverage of nutritional status dimensions, encapsulating physical, emotional, and cognitive health aspects [26]. Furthermore, its internal consistency and reliability have been thoroughly verified in various demographic and clinical contexts, underscoring its significance as an indispensable instrument for the early identification of malnutrition risks among the older people.

### 3.4. Data Collection

Data collection for this study, conducted over a three-month period in 2023, was meticulously organised to ensure a thorough analysis of the nutritional status and feeding behaviours of nursing home residents. The clinical nursing educator, having undergone rigorous training for the study, was responsible for observing participants during mealtimes. These observations were carried out discreetly from a respectful distance, such as across the room or from outside the door, to observe without disrupting the residents’ natural mealtime environment. The collected data included various components, as follows: (i) demographic characteristics, (ii) nutritional status assessed via MNA-SF, and (iii) feeding behaviours documented through both observations and FDI checklist. Initial data collection focused on gathering demographic information of the study participants, which included age, gender, duration of stay in the nursing home, educational level of residents, cognitive status, and the work experience of the nursing staff involved in the study. A paper-based FDI checklist evaluate challenges encountered during mealtime and the MNA-SF to determine the nutritional status of the participants were completed by the clinical nursing educator for each participant during mealtime, detailing observed feeding behaviours. The level of assistance provided to recruited residents, and the portion of food consumed were included in the checklist.

### 3.5. Data Analysis

Data analysis was performed using SPSS version 29 (IBM Corp., Armonk, NY, USA) with a significance level set at 0.05. Descriptive statistics summarised participant demographics, including age, gender, length of stay, antipsychotic use, assistance level, and food intake. The frequencies of the top five feeding difficulties, as measured by the Feeding Difficulty Index (FDI), were calculated.

Spearman’s correlation coefficients were calculated to examine the associations among key variables, including MNA-SF scores (nutritional status), MoCA scores (cognitive function), FDI scores (feeding difficulties), residency duration, food consumption, and level of assistance required. To further explore the predictive relationships, a multiple linear regression analysis was conducted to identify and rank significant predictors of MNA-SF scores. Independent variables in the model included MoCA scores, FDI scores, residency duration, and relevant covariates. Model assumptions, such as multicollinearity and the distribution of residuals were assessed to ensure the validity of the analysis.

### 3.6. Ethical Considerations

This observational study, conducted as part of the ‘Improving nutritional status and mealtime experience in a residential aged care facility through knowledge translation strategy and environmental modifications’ obtained ethics approval from the Human Research Ethics Committee (HREC) at the University overseeing the research (Approval number: H15406; approved on 18 August 2023). Participant information sheets were distributed, and consent forms were signed by the nursing staff and the residents or the next of kin of those who were unable to given consent due to cognitive impairment. Participants were assured of their right to withdraw from the study at any time without any consequences to their care. All data were anonymised and securely stored to protect the participants’ privacy and confidentiality.

## 4. Results

### 4.1. Participant Characteristics

The study involved the collection and analysis of data from 51 residents residing in nursing homes. The data encompassed various parameters such as residents’ gender, age, duration of care, utilisation of antipsychotics, extent of assistance required, food consumption, cognitive and nutritional status. Table 1a,b provides a comprehensive overview of their demographic details. Of these participants, 35 were female and 16 were male. The average age of the participants was 87.8 years, with an age range spanning from 67 to 96 years. A significant portion of the residents (59%) had attained education levels ranging from primary to high school, while 37% held higher education qualifications such as diplomas or bachelor’s degrees. The average duration of residency in the aged care facility was 2.7 years, with the longest duration being 8 years.

Cognitive assessments yielded a mean score of 14, indicating a moderate level of cognitive impairment among participating residents. In terms of severity, 31% of participants demonstrated severe cognitive impairment (scores 0–9), 25% moderate impairment (scores 10–17), and 28% mild impairment (scores 18–25). Only 16% of participants scored within the normal cognitive range (26–31). Additionally, 69% of participants were prescribed antipsychotic medications.

Nutritional assessments using the Mini Nutritional Assessment Short-Form (MNA-SF) revealed an average score of 10.24, suggesting a risk of malnutrition. Specifically, 47.1% of residents were at high risk of malnutrition, and 13.7% were classified as malnourished. Notably, over half of the residents (65%) required assistance with feeding, underscoring their significant dependency on nursing staff support. The typical feeding time lasted approximately 23 min. In terms of food consumption patterns, a majority (67%) consumed three-quarters (3/4) of their meal, while 19% of individuals consumed the entire meal. Additionally, a smaller portion (12%) consumed half (1/2) of their meal, with only 2% consuming one-quarter (1/4) of their meal.

### 4.2. FDI Among Nursing Home Residents

Table 2 shows the top five FDIs, with the most common difficulties being discontinuation of feeding for over 1 min (74.5%), distracted from feeding by talking, looking around, or watching TV (62.8%), does not start to eat for at least 1 min when invited to do so (60.9%), does not open the mouth or bites the utensils when food is offered (51%), and pushes or resists food offered by hand (33.4%).

### 4.3. Correlations Between Residency Duration, Cognitive Function, FDI, Food Consumption, and Nutritional Status

Table 3 presents the correlations among residency duration, cognitive function (MoCA scores), nutritional status (MNA-SF scores), food consumption, and feeding difficulties (FDI scores). A significant positive correlation was found between MoCA and MNA-SF scores (r = 0.401, *p* = 0.037), indicating that cognitive function may be related to nutritional status among residents. Similarly, food consumption was positively correlated with MNA-SF scores (r = 0.392, *p* = 0.004), suggesting that higher dietary intake may be linked to a lower risk of malnutrition.

In contrast, residency duration was negatively correlated with MNA-SF scores (r = −0.292, *p* = 0.037), implying that longer lengths of stay may be associated with poorer nutritional status. Moreover, FDI scores showed a strong negative correlation with MNA-SF scores (r = −0.630, *p* < 0.001), highlighting the significant adverse effect of feeding difficulties on nutritional outcomes.

A multiple linear regression analysis was conducted to examine whether nutritional status predicted the length of stay in the nursing home. The MNA-SF, FDI, and MoCA emerged as the dependent variable entered the model, and MNA-SF emerged as the only significant predictor of the residency duration (Table 4). No other covariates were included in this analysis. The model identified a statistically significant negative association between MNA-SF scores and residency duration (β = −0.299, *p* = 0.033), suggesting that residents with better nutritional status were likely to have shorter lengths of stay. Specifically, each one-point increase in the MNA-SF score was associated with a reduction of approximately 0.34 years in residency duration.

Additionally, scatter plots with polynomial regression lines were generated to visualise the relationships between residency duration and key outcome measures, including MoCA, FDI, and MNA-SF (Figure 1). A downward curvilinear trend was observed between residency duration and MoCA scores, suggesting cognitive decline over time (R^2^ = 0.0374). FDI scores showed a slight upward trajectory with longer stays, indicating increasing feeding difficulties (R^2^ = 0.0822). Similarly, MNA-SF scores displayed a mild downward trend, reflecting a gradual deterioration in nutritional status (R^2^ = 0.0467). Although the effect sizes were modest, the visual patterns support a general progression of cognitive, functional, and nutritional decline among long-term nursing home residents.

### 4.4. Interplay Between Level of Assistance Required, Feeding Difficulties, and Cognitive Function

Table 5 presents the associations between FDI scores and key variables, including cognitive function (MoCA scores), level of assistance required, and food consumption among nursing home residents. FDI scores were significantly negatively correlated with MoCA scores (r = −0.623, *p* < 0.001), indicating that residents with greater feeding difficulties had lower cognitive function. A significant negative correlation was also found between FDI scores and the portion of food consumed (r = −0.392, *p* = 0.004), suggesting that more frequent feeding challenges were associated with reduced dietary intake. Additionally, FDI scores were positively correlated with the level of assistance required (r = 0.763, *p* < 0.001), reflecting that residents experiencing more feeding challenges required higher levels of mealtime support.

## 5. Discussion

### 5.1. Nutritional Status and Cognitive Impairments

The findings revealed that nearly half of the residents were at high risk of malnutrition, with a significant portion already malnourished. These results align with previous research indicating a high prevalence of malnutrition among older people globally [14,27]. An inverse correlation was observed between cognitive function, as measured by MoCA scores and nutritional status.

This association suggests a possible relationship between cognitive impairment and nutritional challenges in older adults, echoing the findings of Bassola et al. (2020), who highlighted the importance of assessing nutritional status alongside cognitive function [6]. However, as the current study is cross-sectional, no causal inference can be drawn.

### 5.2. Feeding Challenges and Support Needs of Cognitively Impaired Residents

Our study reveals that older residents with cognitive impairments in nursing homes face substantial obstacles in food consumption, characterised by diminished food intake, an increased need for mealtime assistance, and a heightened risk of malnutrition. These findings are supported by the research of Lin et al. (2023), which identified similar feeding difficulties due to mealtime distractions, such as talking, looking around, or watching TV [14]. Furthermore, our analysis identified specific behaviours indicative of feeding challenges, including prolonged pauses in feeding and distraction from meals. This aligns with prior research [14] on feeding behaviours in older people across different care settings, suggesting these issues are widespread among the cognitively impaired residents. This pattern accentuates the need for bolstering the competence and insight of caregiving staff to support these residents effectively, enhancing their food consumption and nutritional intake.

Furthermore, our results shed light on the complex interplay between cognitive impairments and the increased vulnerability of nursing home residents to feeding difficulties and malnutrition. This complexity is further delineated by the direct correlation between the FDI scores and the level of assistant required, underscoring the critical role played by nursing staff in mitigating these challenges. The necessity for targeted training programmes emerges, aimed at equipping nursing staff with the skills to identify and manage the distinct feeding difficulties encountered by seniors with cognitive impairments [28]. Such educational initiatives promise not only to elevate nutritional health outcomes but to also significantly improve the life quality of nursing home inhabitants.

Importantly, regression analysis revealed a significant negative association between residency duration and MNA-SF scores, indicating that residents who had stayed longer in the facility tended to experience poorer nutritional outcomes. This trend may reflect the limited continuity of staff training and the absence of ongoing nutrition-focused interventions. As residents’ conditions change over time, especially in the context of advancing cognitive impairment, routine caregiving practices may no longer be sufficient to meet their evolving needs.

This may be due to the fact that, despite numerous studies investigating interventions such as food service and dining environment modifications to improve the nutritional status of people with dementia [29,30], relatively few have focused on enhancing the capacity of nursing staff to assist residents during mealtimes. In many aged care facilities, routine nursing staff nutrition training programmes primarily emphasise fall prevention and personal hygiene [23], while mealtime support remains under-addressed. To bridge this gap, future efforts should consider the development of targeted training programmes that equip staff with the knowledge, skills, and confidence to manage feeding challenges during mealtimes. Such training could play a critical role in promoting safe, effective, and person-centred mealtime care for older people with dementia [6,31].

Although cognitive impairment is a recognised contributor to nutritional decline, the results suggest that organisational factors such as the length of stay and the quality of care delivery may have an equally strong influence. These findings underscore the importance of implementing comprehensive intervention strategies. Key recommendations include ongoing professional development for care staff, the use of individualised mealtime support plans, and the systematic observation and adjustment of resident carer interactions to ensure that nutritional needs continue to be met effectively [32].

Recognising and addressing these interconnected factors are crucial for designing effective interventions aimed at improving the nutritional care and support for older individuals in aged care facilities, ensuring their dietary needs are adequately met and reducing the risk of malnutrition.

### 5.3. Implications for Future Research and Practice

This study underscores several important implications for future research and clinical practice. First, although current dietary strategies in aged care settings align with the Australian Aged Care Quality Standards, malnutrition remains prevalent among older adults with dementia. This highlights the inadequacy of existing approaches in addressing the complex needs of residents experiencing both cognitive decline and feeding difficulties. There is a pressing need for rigorously designed intervention studies aimed at reducing malnutrition and feeding challenges in this vulnerable population. Such studies should aim to deepen our understanding of effective mealtime support strategies and their impact on individuals with dementia [33].

Future research should also explore how improvements to the mealtime environment, including optimal seating arrangements, noise control, and the use of visual cues, can foster more supportive and person-centred dining experiences for residents with cognitive impairment [34]. Furthermore, enhancing staff training on recognising and responding to feeding challenges and risk factors is essential for improving nutritional well-being [28,35]. A more nuanced understanding of both the feeding behaviours of people with dementia and their interactions with nursing staff during mealtimes is also needed.

To address these challenges, future multidisciplinary interventions should involve the development and testing of targeted strategies that promote positive mealtime behaviours and minimise disruptive or resistive ones. These interventions should prioritise practical, scalable approaches, such as personalised assistance, effective communication techniques, and relationship-based, person-centred care models. In addition, interdisciplinary collaboration with dietitians-nutritionists, occupational therapists, and speech therapists should be considered to address not only dietary content but also the mechanics of eating and the overall mealtime environment.

### 5.4. Strengths and Limitations

This study stands out for its comprehensive evaluation of the nutritional status and feeding behaviours among nursing home residents, integrating cognitive function, feeding difficulties, and nutritional assessments through validated instruments like the MNA-SF and FDI. Its multidimensional approach, examining the intricate relationships between cognitive assessments, duration of residency, and levels of assistance, provides deep insights into the multifaceted challenges influencing nutritional health in nursing home. Particularly noteworthy is the focus on residents with cognitive impairments, shedding light on a critical yet often neglected aspect of nutrition in aged care and highlighting the necessity for tailored support strategies.

However, the study is not without its limitations. The smaller than ideal sample size of 51 residents may limit the generalisability of the findings across a broader spectrum of nursing home environments. Additionally, the cross-sectional observational design inherently limits the ability to establish causal relationships between the identified factors and nutritional outcomes. This limitation is partly due to the time-consuming nature of the observational process, as each participant requires approximately 30 min of direct observation during mealtimes. This extended observation period reflects the complexity of capturing real-time behaviours and interactions, which are essential to accurately represent the phenomena but restrict the feasibility of longitudinal or experimental designs within the current study framework. The study’s setting, confined to specific nursing home units willing to participate, also raises questions about the representativeness of the findings across the diverse landscape of aged care facilities. Additionally, the scope of this study did not extend to evaluating how the dining environment or the professional knowledge and feeding skills of the nursing staff impact the nutritional well-being of the residents. These unexplored areas represent significant avenues for future research, offering the potential to deepen our understanding of nutritional care in nursing homes and inform more effective intervention strategies.

Potential sources of bias were considered in the study design. First, selection bias may have occurred, as participation was voluntary and may have attracted staff who were more motivated or had a pre-existing interest in dementia care. Second, self-reported measures were used to assess staff knowledge, attitudes, and confidence, which may be subject to social desirability bias. To reduce these risks, anonymity was maintained during survey completion, and participants were assured that their responses would remain confidential and used solely for research purposes [36].

## 6. Conclusions

In conclusion, this study sheds light on the significant challenges related to nutritional status and feeding behaviours among nursing home residents, particularly those with cognitive impairments. The findings underscore the urgent need for targeted nutritional interventions, enhanced staff training, and a holistic approach to mealtime assistance. Notably, the inclusion of food difficulty as a key focus in the future nursing staff education programme is very necessary, as it allows nursing staff to better recognise and understand to the practical challenges residents face during mealtime. By integrating this concept into nursing staff training, nursing homes can enhance their capacity to deliver person-centred care, ultimately improving residents’ nutritional health, reducing feeding-related behavioural issues, and promoting overall well-being.

## Figures and Tables

**Figure 1 nutrients-17-02607-f001:**
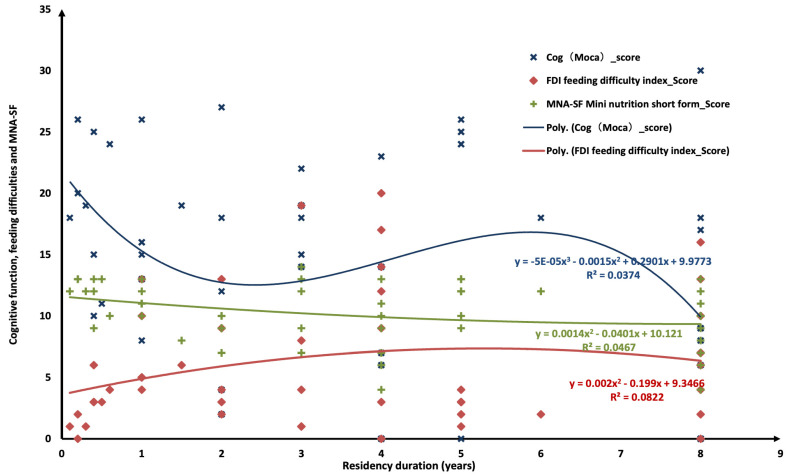
Correlations between residency duration and cognitive function and between feeding difficulties and nutritional status, respectively.

**Table 1 nutrients-17-02607-t001:** Sociodemographic, clinical, nutritional, and (a,b) cognitive characteristics of the study sample (*n* = 51).

**a**				
	**Minimum**	**Maximum**	**Means**	***n* (%)**
Age	67	96	87.8	51
Duration of staying in an aged care facility (years)	0.1	8	2.7	51
Nutritional status				
Feeding time (min)	15	45	23	51
FDI total score	0	20	5.96	51
MNA-SF total score	4	14	10.24	51
Cognitive status (MoCA)	0	30	14	51
**b**				
	** *n* **		**%**	
Gender				
Female	35		69%	
Male	16		31%	
Anti-psychosis medication				
Yes	35		69%	
No	16		31%	
Education				
Not known/stated	2		3.9%	
Primary	12		23.50%	
Secondary school	4		7.8%	
High school	14		27.50%	
Higher education level	19		37.30%	
Cognitive status (MoCA)				
0–9 severe	16		31%	
10–17 moderate	13		25%	
18–25 mild	14		28%	
26–30 normal	8		16%	
MNA-SF score				
0–7 points: malnourished	7		13.7%	
8–11 points: at risk of malnutrition	24		47.10%	
12–14 points: normal nutritional status	20		39.20%	
Level of assistance				
Self-feeding	18		35%	
Partial assistance	18		35%	
Full assistance	15		30%	
Portion of Food Consumed				
1/4	1		2%	
1/2	6		11.80%	
3/4	34		66.70%	
Whole meal	10		19.60%	

Note: *n* = Total number of participants; % = Percentage of participants.

**Table 2 nutrients-17-02607-t002:** Top five most frequent items of FDI (*n* = 51).

Ranking	FDI Item	% Percentage
1	Discontinues feeding for over 1 min	74.50%
2	Distracted from feeding by talking, looking around, or watching TV	62.80%
3	Does not start to eat for at least 1 min when invited to do so	60.90%
4	Does not open the mouth or bites the utensils when food is offered	51.00%
5	Pushes or resists food offered by hand	33.40%

Note: *n* = Total number of participants (51).

**Table 3 nutrients-17-02607-t003:** Correlations of residential duration, MoCA, FDI score, MNA-SF score, and food consumption.

			Residential Duration (Year)	MoCA Score	FDI Score	MNA-SF Score	Food Consumption
**Spearman’s rho**	**Residential Duration (Year)**	Correlation Coefficient	1.000	−0.248	0.181	−0.292 *	−0.299 *
		Sig. (2-tailed)	-	0.079	0.205	0.037	0.033
		*n*	51	51	51	51	51
	**Mo** **CA S** **core**	Correlation Coefficient	−0.248	1.000	−0.623 **	0.401 **	0.162
		Sig. (2-tailed)	0.079	-	<0.001	0.004	0.256
		*n*	51	51	51	51	51
	**FDI Score**	Correlation Coefficient	0.181	−0.623 **	1.000	−0.630 **	−0.392
		Sig. (2-tailed)	0.205	<0.001		<0.001	0.004
		*n*	51	51	51	51	51
	**MNA-SF** **Score**	Correlation Coefficient	−0.292 *	0.401 **	−0.630 **	1.000	0.392 **
		Sig. (2-tailed)	0.037	0.004	<0.001	-	0.004
		*n*	51	51	51	51	51
	**Food Consumption**	Correlation Coefficient	−0.299 *	0.162	−0.392	0.392 **	1.000
		Sig. (2-tailed)	0.033	0.256	0.004	0.004	-
		*n*	51	51	51	51	51

Note: *n* = Total number of participants (51). * *p* < 0.05, statistically significant at the 95% confidence level (2-tailed). ** *p* < 0.01, statistically significant at the 99% confidence level (2-tailed).

**Table 4 nutrients-17-02607-t004:** Multiple linear regression.

Coefficients							
Model		Unstandardized Coefficients			Standardised Coefficients	*t*	Sig.
		B	Std. Error		Beta		
1	(Constant)	6.989	1.613			4.333	<0.001
	MNA-SF Score	−0.335	0.153		−0.299	−2.196	0.033
2	FDI Score	Not significant				Not significant	
3	MoCA Score	Not Insignificant				Not significant	
a. Dependent Variable: Duration in staying in aged care facility (nursing home)							
**Model Summary**							
**Model**	**R**	**R Square**		**Adjusted R Square**		**Std. Error of the Estimate**	
1	0.299a	0.090		0.071		2.63602	
a. Predictors: (Constant), MNA-SF Score							

**Table 5 nutrients-17-02607-t005:** Correlations of FDI/MoCA/level of assistance.

		Portion of Food Consumed	The Score of Cognitive Assessment (MoCA)	Level of Assistance
**FDI score**	Spearman’s coefficient correlation	−0.392 *	−0.623 **	0.763 **
	Sig. (2-tailed)	0.004	<0.001	<0.001
	*n*	51	51	51

Note: *n* = Total number of participants (51). * *p* < 0.05, statistically significant at the 95% confidence level (2-tailed). ** *p* < 0.01, statistically significant at the 99% confidence level (2-tailed).

## Data Availability

The data are available on request from the corresponding author. The reason for restriction is that the dataset contains personal information, so it cannot be made publicly available. The corresponding author will assess each request and determine which data can be provided in compliance with ethical guidelines and institutional requirements.

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
