# Peer review of "Nutritional Status and Feeding Difficulty of Older People Residing in Nursing Homes: A Cross-Sectional Observational Study"

_nutrients, 2025, doi:10.3390/nu17162607_

Round 1
Reviewer 1 Report
Comments and Suggestions for Authors
STRUCTURE
- The manuscript is correctly structured. The comments are taken into account, but they still need to improve on these two aspects prior to publication:
- Table 1. Characteristics of the study sample(N=51).
- Include below each table the meaning of the abbreviations used, such as % and N. Applicable to the rest of the document.
- There are data that are not provided in the table. Please separate the variables into 2 tables, so that the boxes are filled in with the data you provide.
- Table 3. Table 3: When a table is divided into two sheets, the sections must be reinserted in the header. Applicable to the rest of the manuscript.
- The section above the introduction highlights and points out: the implications for the profession and/or patient care, the impact, compliance with the STROBE guidelines, the population involved, and the contributions of this article to the global clinical community. However, in accordance with the journal's guidelines, authors are recommended to distribute these characteristics in their corresponding sections: For example:
- Abstract: Brief statement on clinical relevance or potential impact.
- Introduction: The population involved (why it is relevant to study university students); The knowledge gap that this study addresses; Brief mention of the expected clinical or social impact.
- Discussion: How these results may influence clinical practice; Possible implications for public policy, prevention, or health education; Relevance to the international scientific community.
- Conclusions: A call to action or a message to the clinical community.
- Materials and methods: Mention STROBE compliance at the end of the methodology section. “This study followed the STROBE guidelines for reporting cross-sectional studies.”
- Some journals even allow you to include the STROBE checklist as a supplementary file, which I recommend if it is Nutrients or another MDPI journal.
- Check the journal's guidelines
- https://www.mdpi.com/journal/nutrients/instructions
- Table 1. Characteristics of the study sample(N=51).
TITLE AND ABSTRACT
- Study design identified in the title: clearly indicates observational, but it is recommended that the title be modified by noting “Cross-Sectional Study”.
- Informative and balanced summary: Provides objectives, methods, main results, and conclusions.
- The title is clear, concise and specific. It adequately communicates the relationship between Mediterranean diet and food addiction in college students.
- The abstract complies with the structured format (Background, Methods, Results, Conclusion) required by Nutrients.
- However, the “Results” section of the abstract could specify correlation coefficients or key data (e.g., r, p value).
INTRODUCTION
- “In our understanding of their feeding…”. Remember not to speak in the first-person plural.
- The introduction is well structured, with solid references (PubMed, 2020-2023). But the section lacks a more solid and specific discussion of the type of diet recommended in the context of geriatric residential care, especially for older adults with cognitive impairment. Although there is a general mention of malnutrition and eating problems in nursing homes, it is not contextualized with current evidence, nor are the most appropriate dietary models for this vulnerable group mentioned.
- It is suggested that this section be strengthened by incorporating references and arguments on the optimal nutritional characteristics for older adults in institutionalized settings, considering:
- The importance of a nutritionally dense diet (rich in high biological value proteins, healthy fats, and micronutrients such as vitamin D, calcium, and B vitamins).
- The proven benefits of dietary patterns such as the adapted Mediterranean diet, which has shown positive effects on cognitive function, inflammatory status, and cardiovascular health in older adults.
- The need to adapt food texture (texture-modified or pureed foods) in cases of dysphagia, which is common in this population.
- The relevance of the psychosocial and structural environment of meals, such as meal times, assistance with eating, and plate presentation, which significantly impact effective intake in institutionalized older adults.
- The consideration of personalized nutritional strategies, such as oral supplementation or energy-protein enrichment of preparations, in patients with low.
- On the other hand, in geriatrics, the clinical term “Age-related malnutrition syndrome” can be used, which includes sarcopenia, dysphagia, anorexia of aging, frailty, and functional loss. Since it can be a symptom within other disorders, such as dementia or depression. For medical and diagnostic coding, ICD-10 or ICD-11 is mainly used, with specific codes for dysphagia, malnutrition, sarcopenia, etc.
- Although the problem is mentioned, the hypothesis of the study is not explicitly stated, and could be strengthened with a clear statement of the type:
- “It is hypothesized that a lower adherence to the Mediterranean diet is associated with a higher prevalence of food addiction in Spanish university students.”
METHODS
- Study design described at baseline (item 4)
- Description of setting, dates, and participants (items 5-6)
- Definition of variables, exposure, outcomes, and measurement tools (items 7-8)
- Managing bias and sample size (items 9-10)
- Well-defined statistical methods (item 12)
- Aspects to be improved:
- Review citation: Kaiser et al. (2009).
|
Point |
Observation |
Recommendation |
|
Sample size |
No justification is provided on how the sample size was determined |
Add a power analysis or empirical justification for the sample size |
|
Potential bias |
Selection or response biases are not discussed |
Include potential limitations due to participant self-selection |
|
Measurement |
The use of the FDI is mentioned, but the diagnostic “cutoff” is not detailed |
Specify how "food difficulty" was defined |
RESULTS
- Data are clearly presented in well-organized tables (age, sex, BMI, Mini Nutritional Assessment Short-Form (MNA-SF) scores and Feeding Difficulty Index (FDI)
- Percentages of students with high adherence to the Mediterranean diet and prevalence of AF are correctly reported
- Spearman correlation analysis, Mann-Whitney U test and multivariate logistic regressions were used
- Recommendations:
- Better explain which variables were included as covariates in the regressions.
- Include confidence intervals (95%CI) for all estimates.
- In Table 2, it would be useful to add an “effect” column (OR or beta, depending on analysis).
- Table 3. It is not necessary to include definitions of abbreviations used, such as Feeding difficulty index, in the table.
DISCUSSION
- The discussion presents the key findings, limitations addressed (item 19) and generalization of the results (item 21). It also cautiously interprets the findings and results are compared with previous studies (both European and international).
- This section lacks specific recommendations for practical implementation in nursing homes:
- Although the study outlines the factors associated with malnutrition and eating difficulties in older adults with cognitive impairment, there is no structured intervention proposal or practical guide for immediate application in residential settings. The research identifies key problems but does not offer specific dietary recommendations, guidelines for adapting the food environment, or examples of menus suitable for this population. Therefore, it is recommended that the authors include or briefly discuss the characteristics of an appropriate diet for institutionalized older adults, such as:
- Modified texture (soft, pureed, or adapted diets according to swallowing ability).
- Sufficient calorie and protein intake (high-calorie/high-protein diet if there is a risk of malnutrition).
- Nutritional enrichment (adding powdered milk, olive oil, creams).
- Frequency and fractionation (more small meals).
- Incorporation of foods with high nutritional density.
- Cultural adaptation of food according to residents' preferences and context.
- Adequate hydration through appealing and easily accessible liquids.
- In addition, it would be valuable to mention interdisciplinary intervention strategies, such as collaboration with dietitians-nutritionists, occupational therapists, and speech therapists, to address not only dietary content but also the mechanics of eating and the experience of the environment.
- Nevertheless, some suggestions for improvement are offered. The limitations section is somewhat terse: it should be expanded to include:
- Self-report bias (questionnaires).
- Lack of national representativeness.
- Impossibility of establishing causality (due to the cross-sectional design).
CONCLUSIONS
- Relevance of including “food difficulty” in food education programs.
Author Response
Thank you for your valuable feedback. We truly appreciate it. For more details, please refer to the attached response table.

Reviewer 2 Report
Comments and Suggestions for Authors
Ref.: nutrients-3786107
This is a cross-sectional study aiming to investigate the feeding behaviors and nutritional status of nursing home residents as well as the impact of cognitive disability and feeding difficulty on nutritional health. Participants were old aged with moderate degree of cognitive dysfunction. The authors observed that almost half of them were at least at high risk for malnutrition and more than half required assistance with feeding. More severe cognitive dysfunction and longer duration of nursing home residency were associated with poorer nutritional outcomes. This observational study adds to the literature, by supporting the notion that cognitive impairment and feeding difficulties affect significantly the nutritional status of nursing home residents, leading to malnutrition and necessitating mealtime assistance.
Since the aging population and concomitantly the prevalence of dementia is rising worldwide, the need for nursing home placement also increases. In this setting, maintaining a good nutritional status for such old-aged residents may be necessary in order to ensure better care outcomes, but may become a significant challenge. Thus the present study is welcomed.
The design is well-designed and executed. The tools (scales) used are simple, yet robust and more than adequate. Statistics are appropriate and the tables are informative. The authors discuss the strengths and limitations of the study (including the small sample size).
One point
The population studied is a mixed population. Most have some degree of cognitive impairment, but neither the cause (Alzheimer’s disease, vascular cognitive impairment, Lewy body dementia) was studied, nor the presence or cause of possible movement disability including parkinsonism, pseudobulbar palsy etc. Different causes and clinical pictures and some treatments may have different effects in appetite, feeding difficulties (possible dysphagia) and patient cooperation during feeding assistance. Of course, subgroup analysis cannot be performed in this low sample size (n = 51) and was not in the aim of this study, but a short discussion on these parameters could be added in the limitations subsection, or in the subsection for implications for future research.
Author Response

(The authors gave the same response as above.)
